# Phytomediated stress modulates antioxidant status, induces overexpression of *CYP6M2*, *Hsp70*, *α-esterase*, and suppresses the *ABC transporter* in *Anopheles gambiae* (*sensu stricto*) exposed to *Ocimum tenuiflorum* extracts

Harun K. Aremu[1,2]*, Christianah A. Dare[1], Idris A. Adekale[1], Bukunmi D. Adetunji[1], Dickson A. Musa[3,2], Luqmon A. Azeez[4], Olu I. Oyewole[1]

**1** Department of Biochemistry, Osun State University, Osogbo, Nigeria, **2** Trans-Saharan Disease Research Centre, Ibrahim Badamasi Babangida University, Nigeria, **3** Department of Biochemistry and Biotechnology, Ibrahim Badamasi Babangida University, Nigeria, **4** Department of Pure and Applied Chemistry, Osun State University, Osogbo, Nigeria

* kolawole.aremu@uniosun.edu.ng

## Abstract

The incorporation of phytoactive compounds in the management of malarial vectors holds promise for the development of innovative and efficient alternatives. Nevertheless, the molecular and physiological responses that these bioactive substances induce remain underexplored. This present study investigated the toxicity of different concentrations of aqueous and methanol extracts of *Ocimum tenuiflorum* against larvae of *Anopheles gambiae* (*sensu stricto*) and unraveled the possible underlying molecular pathways responsible for the observed physiological effects. FTIR and GCMS analyses of phytoactive compounds in aqueous and methanol crude extracts of *O. tenuiflorum* showed the presence of OH stretching vibration, C = C stretching modes of aromatics and methylene rocking vibration; ring deformation mode with high levels of trans-β-ocimene, 3,7-dimethyl-1,3,6-octatriene in aqueous extract and 4-methoxy-benzaldehyde, 1,3,5-trimethyl-cyclohexane and o-cymene in methanol extract. The percentage mortality upon exposure to methanol and aqueous extracts of *O. tenuiflorum* were 21.1% and 26.1% at 24 h, 27.8% and 36.1% at 48 h and 36.1% and 45% at 72 h respectively. Using reverse transcription quantitative polymerase chain reaction (RT-qPCR), down-regulation of *ABC transporter*, overexpression of *CYP6M2*, *Hsp70*, and *α-esterase*, coupled with significantly increased levels of SOD, CAT, and GSH, were observed in *An. gambiae* (*s.s.*) exposed to aqueous and methanol extracts of *O. tenuiflorum* as compared to the control. Findings from this study have significant implications for our understanding of how *An. gambiae* (*s.s.*) larvae detoxify phytoactive compounds.

**Data Availability Statement:** All relevant data are within the manuscript and its Supporting information files.

**Funding:** The author(s) received no specific funding for this work.

**Competing interests:** The authors have declared that no competing interests exist.

## Introduction

In sub-Saharan Africa, the mosquito species *Anopheles gambiae* (*sensu stricto*) is highly effective at transmitting malaria and remains a substantial public health threat in the region. Projections indicate that Sub-Saharan Africa will face an estimated 228 million malaria cases and 602,000 malaria-related fatalities [1]. Considering the extensive distribution of *An. gambiae* (*s.s.*) in tropical and sub-tropical areas, an estimated fifty percent of the world's populace is vulnerable to *Plasmodium falciparum* infections transmitted by this species of mosquito. Consequently, controlling mosquito populations is the most effective approach to reducing infection rates [2].

Mosquitoes defend themselves against plants and external toxins by utilizing protein systems that neutralize toxins. A large number of genes that encode pumps for efflux and detoxification enzymes create these systems. They either alter the harmful chemicals or eliminate them from cells. Insects utilize metabolic pathways to remove xenobiotics through a series of mechanisms that involve detoxification enzymes and transporters. The gene superfamilies of cytochrome P450s and esterases have been extensively studied and are known to have a crucial role in the process of detoxification [3, 4]. Cytochrome P450 enzymes play a vital role in the detoxification process, particularly in the metabolism of both synthetic and natural compounds. The upregulation of the CYP450 genes could influence the properties of endogenous molecules or cellular products, potentially altering their harmful features. According to Vivekanandhan et al. [5], the production of CYP450s and their enzyme functions in mosquitoes may help them adapt to changing conditions and become resistant to pesticides.

David et al. [6] found that the *CYP6M2* gene class plays a role in partially causing pyrethroid resistance in *An. gambiae*. Specific families and/or members within these superfamilies have greater efficiency in detoxifying particular xenobiotics and pesticides compared to others [7]. Furthermore, studies have linked heat shock proteins (HSPs) and ATP-binding cassette (ABC) transporters to the process of insect detoxification. However, Traverso et al. [8] noted that these specific members of the superfamily had not received much research. ABC transporters are crucial for cell defense since they facilitate the removal of toxicants from cells. According to Dermauw and Van Leeuwen [9], ABC transporters play a significant role in protecting mosquitoes from various types of insecticides. Insecticide exposure elevates the activity of ABC transporter genes in *Anopheles* larvae from susceptible populations. This leads to a reduction in the intracellular concentration of toxicants. This supports the idea that these transporters are very important for the mosquito's defense systems [10]. HSPs, which work as molecular chaperones to maintain the proper conformation of proteins, are also capable of responding to environmental stress. In response to stress, various insect studies have shown that certain heat shock proteins, such as Hsp20, Hsp70, and Hsp90, are expressed [11–13].

There has been a resurgence of interest in finding new plant-based mosquito control options that are less detrimental to the environment in an effort to find alternatives to synthetic insecticides [14]. Research on bioactive phytocompounds has been extensive, but most of these compounds still lack information about their molecular mechanisms of action. This information could be useful for developing insecticides that are selective, target-specific, and resistant to malaria-causing vectors. Divekar et al. [15] found that some plant compounds are directly toxic or limit growth, while others impart repelling properties.

Holy Basil (*Ocimum tenuiflorum*) is a plant belonging to the Lamiaceae family. It is well known for its medicinal benefits and is one of the oldest plants in the Ocimum genus. The medicinal and curative activities of the phytochemicals present in *O. tenuiflorum* have been extensively studied. The plant has high levels of flavonoids and potent antioxidants, which can help protect against oxidative damage. Ragavendran et al. [16] and Bhavya et al. [17] reported

the insecticidal properties of *O. tenuiflorum* essential oil, with its ability to suppress acetylcholinesterase activity. It has also been reported that *O. tenuiflorum* L. is an abundant source of essential oils, including 1-methyl-3-(1′-methylcyclopropyl) cyclopentene, phenol, and 2-methoxy-4-(1-propenyl), 1,3-isobenzofurandione, 3,7,11,15-tetramethylhexadeca-1,3,6,10,14-pentaene, and 3a,4,7,7a-tetrahydro-4,7-dimethyl, recognized for their ability to induce a mosquitocidal response in various species of mosquitoes, such as *Cx. quinquefasciatus*, *An. stephensi*, and *Ae. aegypti* [16]. Although the insecticidal properties of *O. tenuiflorum* extracts are well documented, there is less understanding of the underlying molecular pathways responsible for the observed adverse physiological effects. Additionally, the extraction solvent significantly influences the bioavailability and characteristics of phytochemicals [14]. Therefore, this study characterized the phytoactive constituents of *O. tenuiflorum* using Fourier Transform Infrared Spectroscopy (FTIR) and Gas chromatography mass spectroscopy (GCMS) analyses. Also, reverse transcription quantitative polymerase chain reaction (RT-qPCR) was used to investigate the molecular mechanisms of action of these compounds and the change in gene expression profiles of *CYP6M2*, *Hsp70*, *α-esterase*, and the *ABC transporter*. Our results indicate that both aqueous and methanol extracts of *O. tenuiflorum* demonstrate a moderate lethal effect on *An. gambiae* (*s.s.*) larvae. Furthermore, exposing the larvae to the phytoactive components in these extracts leads to a multifaceted physiological response, including the activation of the detoxification process through the overexpression of *CYP6M2*, *Hsp70*, and *α-esterase*, along with elevated levels of antioxidant enzymes. Importantly, the potential down-regulation of the *ABC transporter* marks the observed toxicity characteristics of the extracts.

## Materials and methods

### Ethical statement

None of the experiments utilized endangered plant species, and no specific authorizations were required to acquire samples. However, consent was obtained from the Agbekoya Farmers Association for the utilization of the plant.

**Study site and mosquito larvae samples.** The larvae of *An. gambiae* (*s.s.*) were collected in April 2023 from Old-Garage, Osogbo, Nigeria. The larvae were transported to the laboratory at the Department of Biochemistry, Osun State University under standard insectary conditions (25˚C to 30˚C, and pH range of 6.97 to 7.04) in a plastic containing distilled water and subsequently identified morphologically. Following the emergence of fourth-instar larvae, they were subjected to bioassay screening.

**Plant collection and extraction.** The *O. tenuiflorum* leaves (Fig 1) were collected from the Oke-Baale area in Osogbo, Osun State (7˚46'05.9˚N, 4˚35'57.0˚E) and morphologically identified by a botanist in the Department of Plant Biology at Osun State University, Nigeria. The leaves were rinsed with purified water, spread out, and let dry at ambient temperature for three weeks. Subsequently, the dried leaves were pulverized into small particles to enhance the surface area for the extraction of phytochemicals. These particles were then placed in a tightly sealed container and maintained at room temperature. The extraction process was conducted individually using aqueous and methanol solvents. Using the modified approach of Muema et al. [18], maceration was performed by adding 500 g of powdered *O. tenuiflorum* leaves with 2 L of 90% methanol and distilled water separately. The mixture was allowed to stand for 72 h with continuous agitation to facilitate sufficient extraction. Subsequently, the extract obtained from each of the solvent extraction methods underwent filtration using Whatman 1 filter paper (Whatman Inc., Haverhill, USA) to eliminate any undesired residue. The crude extract's filtrate was evaporated until it became dry and then stored in an airtight container for further use.

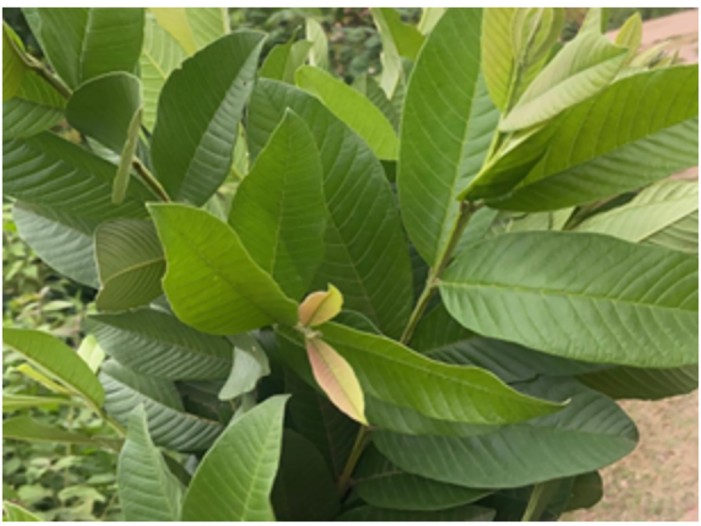

**Fig 1. *Ocimum tenuiflorum*.**

**FTIR and GCMS analyses of extracts of *O. tenuiflorum*.** The solvent crude extracts were analyzed by FTIR to identify the individual bioactive components and their molecular structures. The spectra collected ranged between 400 and 4000 cm$^{-1}$ using a SHIMADZU FTIR-8400s spectrophotometer. Also, the Shimadzu GCMS equipment was used for the GC-MS analysis of both the methanol and aqueous crude extracts. The analysis was conducted on a Shimadzu GCMS-Ultra system with an Agilent 19091J-413 capillary column (30 m × 320 m x 0.25 m; maximum temperature, 370°C) for separation. The carrier gas was ultra-high-quality helium (99.99%) and the flow rate was kept constant at 1.4871 mL/min while the pressure was held steady at 1.4902 psi., temperature of 280°C and kept in the injection line, transfer line, and ion source. The oven temperature was set to rise from 80°C (maintained for 2 min) to 280°C at a rate of 3°C per min. The data was gathered by recording full-scan mass spectra in the range of 40–550 amu. The percentage based on peak area was calculated which helped to establish the relative composition of the elements inside the extracts. Chemical components within the various solvent extracts were identified and characterized using GC retention time. The NIST14 mass spectral libraries were used to perform a computational mass spectral match.

**Bioassay test.** Larval bioassays were conducted comparatively on *An. gambiae* (*s.s.*) larvae exposed to aqueous and methanol crude extracts. Bioassays were performed in triplicate with 20 larvae in 100 mL solution. The test concentrations of 12.5, 25, 50, 125, and 250 mg/L were prepared by dissolving the dried crude extracts of *O. tenuiflorum* in 1 mL of ethanol and then making the volume up to 100 mL. A mixture of 99% water and 1% ethanol was used as a control. Larval mortality was monitored after 24, 48 and 72 h contact with the test solutions. Any larvae that showed no signs of movement and were unable to rise to the surface were considered dead. Differential tolerance levels between the larvae exposed to each concentration and controls were further analyzed by a Generalized Linear Model (GLM) from dose-mortality data.

**Antioxidant biomarkers assay.** The larvae were homogenized and centrifuged at 4°C for 20 min at 3500 rpm in 20 mM ice-cold sodium phosphate buffer (pH 7.0). The supernatant was monitored for the activities of superoxide dismutase (SOD), glutathione peroxidase (GPx), catalase (CAT), and glutathione (GSH) concentrations, as described by Mistral and Fridovich [19], Rotruck et al. [20], and Claiborne [21], Jollow et al. [22] respectively.

**Expression profile of Hsp70, ABC transporter, CYP6M2 and α-esterase.** The expression profile of *Hsp70*, *ABC transporter*, *CYP6M2*, and *α-esterase* in *An. gambiae* (*s.s.*) larvae were analyzed using RT-qPCRs after 72 hr of exposure to methanol (Group 1) and aqueous (Group 2) extracts of *O. tenuiflorum*. RNA extraction was performed for each group of treated larvae according to the instructions provided by Qiagen, using their RNeasy Mini Kit (Hilden, Germany). Subsequently, the RNA was isolated using a solution free of enzymes and its concentration was determined at a wavelength of 260 nm using QubitTM fluorometric quantitation. Using a QuantiTect Reverse Transcription Kit (Qiagen, Hilden, Germany) together with random hexamers, cDNAs were generated from 150 ng of total RNA. The primers employed in RT-qPCRs, derived from the cDNA template, are highlighted in Table 1. The RT-qPCRs were performed on target genes using a BioRad CFX Real-Time PCR Detection System (Bio-Rad, California, USA) under the following conditions: 10 μl of qPCR Master Mix, 0.5 μl of forward and reverse primers, 95°C for 1 min, with 40 cycles of amplification at 95°C for 15 sec and 60°C for 1 min. After each cycle, fluorescence signals were measured from duplicated reactions. The cycling threshold (Ct) values were determined for each run by calculating the mean and subsequently normalizing them using the endogenous reference gene α-tubulin. The 7500 Software v2.0, (Thermo Fisher Scientific in Massachusetts, USA), was utilized for data processing. Amplification plots were constructed by plotting the variance in linear amplification of fluorescence (ΔRn) against the number of cycles. The values of Means and SD of Ct values were obtained for each duplicate reaction. The fold changes in expression levels were calculated using $2^{-\Delta\Delta Ct}$ method [23].

**Statistical analysis.** The data were expressed as the mean ± standard deviation (n = 3). One-way analysis of variance (ANOVA) with Duncan test was used to analyze normally distributed data, while Kruskal-Wallis test was employed for non-normal data. A Kaplan-Meier survival analysis was performed to estimate the differential tolerance level between the test crude extracts. $P < 0.05$ was considered statistically significant. Graphs were designed using GraphPad Prism 9.5.1 for Windows (GraphPad Software, San Diego, USA).

## Results

### Characterization of bioactive compounds in crude extracts of *O. tenuiflorum*

FTIR and GCMS analyses of phytoactive compounds in aqueous and methanol crude extracts of *O. tenuiflorum* are represented in Figs 2 and 3 respectively. The FTIR spectra revealed

**Table 1. RT-qPCR gene-specific primer.**

| Gene | Sequence (5'– 3') | Amplicon size | Reference |
|---|---|---|---|
| α-tubulin (internal reference) | F: CAATGAGGCGATCTACGACA | 171 bp | Muema et al. [18] |
| | R: TACGGCACCAGATTGGTCT | | |
| *ABC transporter* | F: TGAGATGTAGGCACTTGAACTAT | 185 bp | Epis et al. [10] |
| | R: CTGTCACCGTTCCACCTAT | | |
| *Hsp70* | F: ACGCCAACGGTATTCTGAAC | 197 bp | Muema et al. [18] |
| | R: ACAGTACGCCTCGAGCTGAT | | |
| *CYP6M2* | F: AGGTGAGGAGAGTCGACGAA | 235 bp | Muema et al. [18] |
| | R: ATGACACAAACCGACAAGG | | |
| *α-esterase* | F: CATCATACCGTCGTTTGTCG | 540 bp | Vivekanandhan et al. [5] |
| | R: GCTTGAGGGTTTGCTTTCAG | | |

F: Forward primer; R: Reverse primer

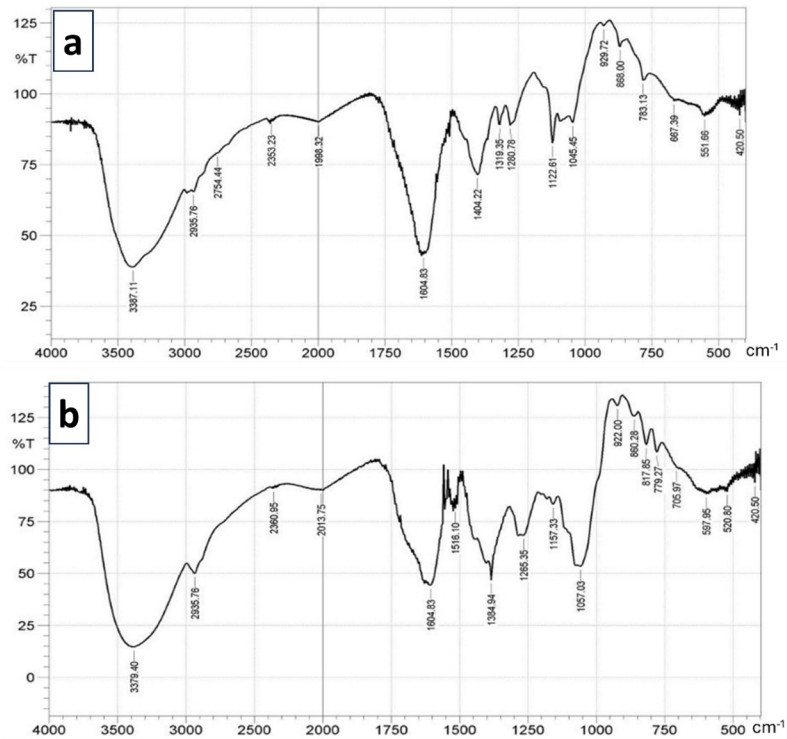

**Fig 2. FTIR Spectra of extracts of *O. tenuiflorum*.** (A) and (B) show FTIR spectra of aqueous and methanol extracts respectively.

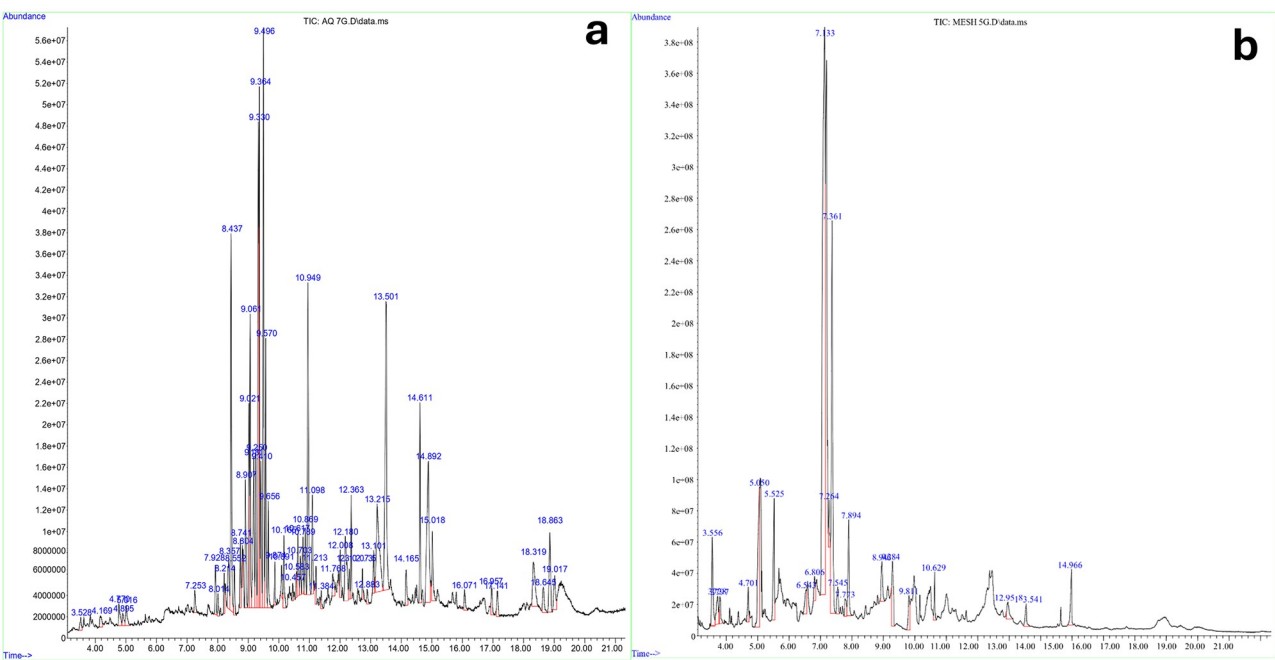

**Fig 3. GC-MS analysis of bioactive constituents of *O. tenuiflorum*.** (A) and (B) show the presence of peaks of prominent bioactive compounds in extracts of aqueous and methanol extracts respectively.

**Table 2. Bioactive constituents of crude extracts of *O. tenuiflorum*.**

| S/N | Peak | R$_t$ (min.) | Peak Area (%) | Name of Compound | Study |
|-----|------|------|------|------|------|
| **Aqueous extract** | | | | | |
| 1 | 11 | 8.437 | 5.75 | trans-β-Ocimene | Chaaban et al. [25] |
| 2 | 16 | 9.021 | 2.55 | 6-methyl-2-methylene-6-(4-methyl-3-pentenyl)-Bicyclo[3.1.1]heptane | Karthi et al. [26] |
| 3 | 17 | 9.061 | 3.63 | 1-(1,5-dimethyl-4-hexenyl)-4-methyl-Benzene | Hamada et al. [27] |
| 4 | 18 | 9.181 | 2.84 | 5-(1,5-dimethyl-4-hexenyl)-2-methyl-1,3-Cyclohexadiene | Almadiy et al. [28] |
| 5 | 19 | 9.250 | 2.32 | cis-α-Bisabolene | Alimi et al. [29] |
| 6 | 20 | 9.330 | 7.43 | 3,7-dimethyl-1,3,6-Octatriene | Srivastava et al. [30] |
| 7 | 21 | 9.364 | 4.04 | p-Cymene | Gong and Ren [31] |
| 8 | 22 | 9.410 | 2.16 | α-Farnesene | Chaaban et al. [25] |
| 9 | 47 | 13.215 | 4.11 | Hexadecyl-Oxirane | Mathivanan et al. [32] |
| 10 | 48 | 13.501 | 7.46 | Octadecanoic acid | Arunthirumeni et al. [33] |
| 11 | 51 | 14.892 | 5.02 | cis,cis,cis-7,10,13-Hexadecatrienal | Fatema et al. [34] |
| **Methanol extract** | | | | | |
| 12 | 1 | 3.556 | 3.80 | o-Cymene | Brandão et al. [35] |
| 13 | 5 | 5.050 | 7.10 | 1,3,5-trimethyl-Cyclohexane | Bezerra-Silva et al. [36] |
| 14 | 9 | 7.133 | 44.80 | 4-methoxy-Benzaldehyde | Chu et al. [37] |
| 15 | 14 | 7.894 | 3.54 | α-methyl-Benzenemethanol | Sanei-Dehkordi et al. [38] |

similar peaks at 3387–3379 cm$^{-1}$ (OH stretching vibration), 2935–2754 cm$^{-1}$ (C-H stretching vibrations corresponding to CH$_3$ and asymmetric stretching of C-H corresponding to CH$_2$), 1998–1604 cm$^{-1}$ (C = C stretching modes of aromatics), 1404–1384 cm$^{-1}$ (C = O of aldehyde group), 1057–1045 cm$^{-1}$ (stretching of C = O), 922–817 cm$^{-1}$ (OH group), 783–779 cm$^{-1}$ (methylene rocking vibration; ring deformation mode) [24]. The GC-MS analysis of aqueous, and methanol crude extracts of *O. tenuiflorum* showed the presence of 11 and 4 bioactive compounds respectively with a percentage peak area greater than 2.0. The 4 prominent compounds with highest peak identified in each of the solvent extracts of *O. tenuiflorum* are: aqueous extracts; cis,cis,cis-7,10,13-hexadecatrienal (5.02%), trans-β-ocimene (5.75%), 3,7-dimethyl-1,3,6-octatriene (7.43%), octadecanoic acid (7.46%); methanol extract; 4-methoxy-benzaldehyde (44.80%), 1,3,5-trimethyl-cyclohexane (7.10%), o-cymene (3.80%), α-methyl-benzene-methanol (3.54%) (Table 2).

## Larvae biotoxicity assay

The percentage mortality of larvae of *An. gambiae* (*s.s.*) at 24, 48 and 72 h induced as a result of exposure to different concentrations of aqueous and methanol extracts of *O. tenuiflorum* is investigated (Fig 4). The percentage mortality upon exposure significantly varies with time. The highest mortality caused due to exposure to different concentrations of methanol and aqueous extracts were 21.1% and 26.1% at 24 h, 27.8% and 36.1% at 48 h and 36.1% and 45% at 72 h respectively. Similarly, the survival rates of larvae exposed to different concentrations of aqueous and methanol extracts of *O. tenuiflorum* relative to the control were evaluated. Concentration-dependent survival rates were observed with high survival rates shown at 12.5 mg/L, and the least rate observed at 250 mg/L (Fig 5).

## Differential gene expression profile

The molecular basis of the stress tolerance of larvae of *An. gambiae* (*s.s.*) treated with methanol and aqueous extracts of *O. tenuiflorum* is investigated by profiling the gene expression

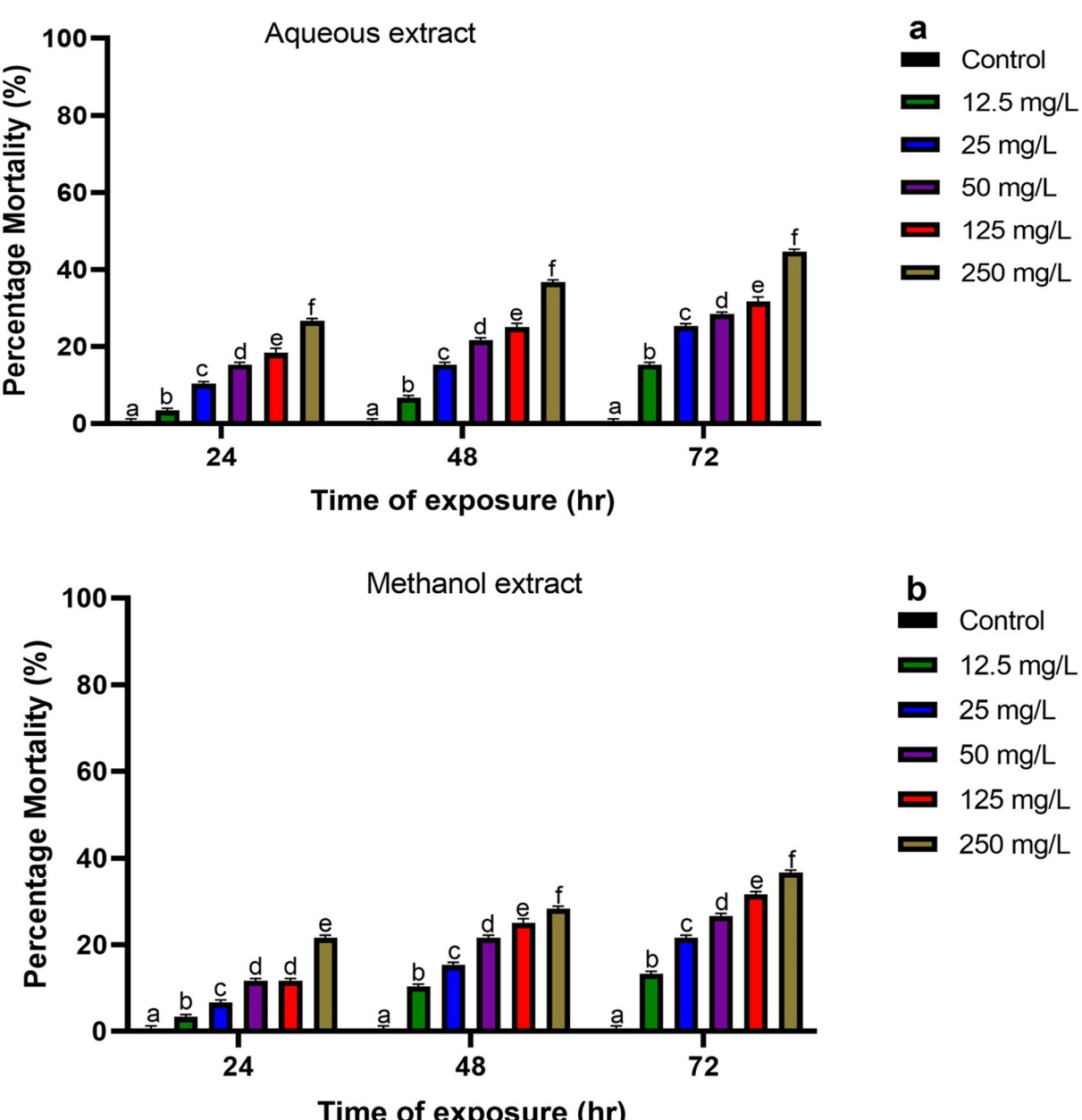

**Fig 4. Percentage mortality at 24, 48 and 72 h of larvae of *An. gambiae (s.s.)* treated with extracts of *O. tenuiflorum*.** Exposure of *An. gambiae* (*s.s.*) larvae to aqueous (A) and methanol (B) extracts of *O. tenuiflorum* caused moderate mortality. Results are given as a mean ± S.D of triplicates. Different superscripts ($^{a,b,c,d,e,f}$) on the bar indicate significant difference at p < 0.05 (ANOVA).

of *CYP6M2*, *Hsp70*, *α-esterase* and *ABC transporter* at 72 h (Fig 6). There was a significant increase in the expression of the selected genes in treated larvae relative to the control (p < 0.05). For instance, in larvae treated with methanol (Group 1) and aqueous (Group 2) extracts of *O. tenuiflorum*, expression of *CYP6M2* increased by 17-fold and 16-fold, *Hsp70* increased by 26-fold and 41-fold, α-esterase increased by eightfold, and 23-fold respectively as compared to the control. Notably, the expression of *ABC transporter* was significantly

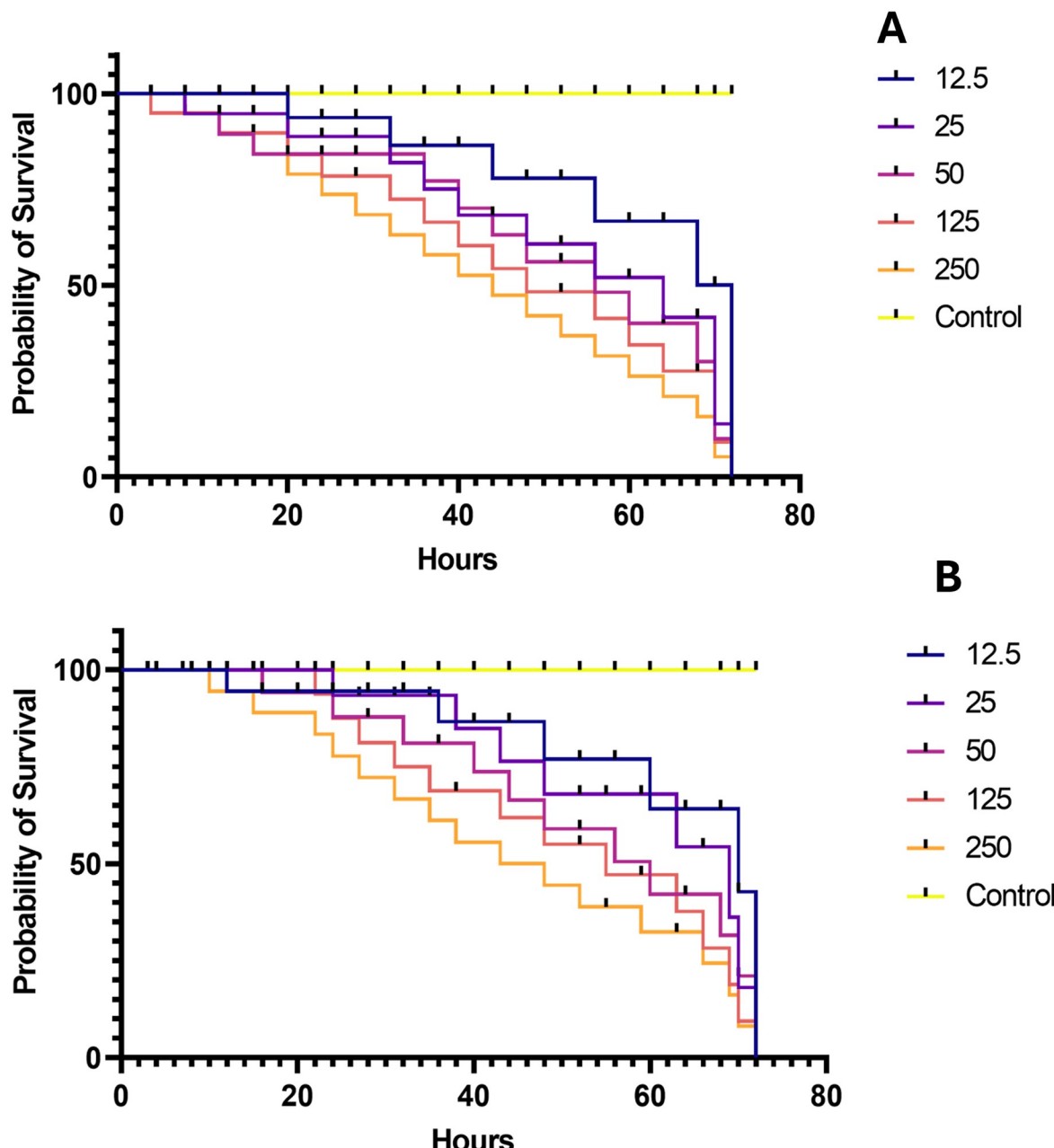

**Fig 5.** Survival rates of larvae exposed to different concentrations of aqueous (A) and methanol (B) extracts of *O. tenuiflorum*. Survival rates were observed to be concentration-dependent with 12.5 mg/L and 250 mg/L of each extracts showing highest and least survival rates respectively.

downregulated in larvae treated with the aqueous and methanol extracts as compared to the control ($p < 0.05$). Furthermore, *Hsp70*, *α-esterase* and *ABC transporter* were differentially expressed between the larvae of *An. gambiae (s.s.)* treated with methanol and aqueous extracts of *O. tenuiflorum* while the transcript encoding the *CYP6M2* were not differentially expressed ($p > 0.05$).

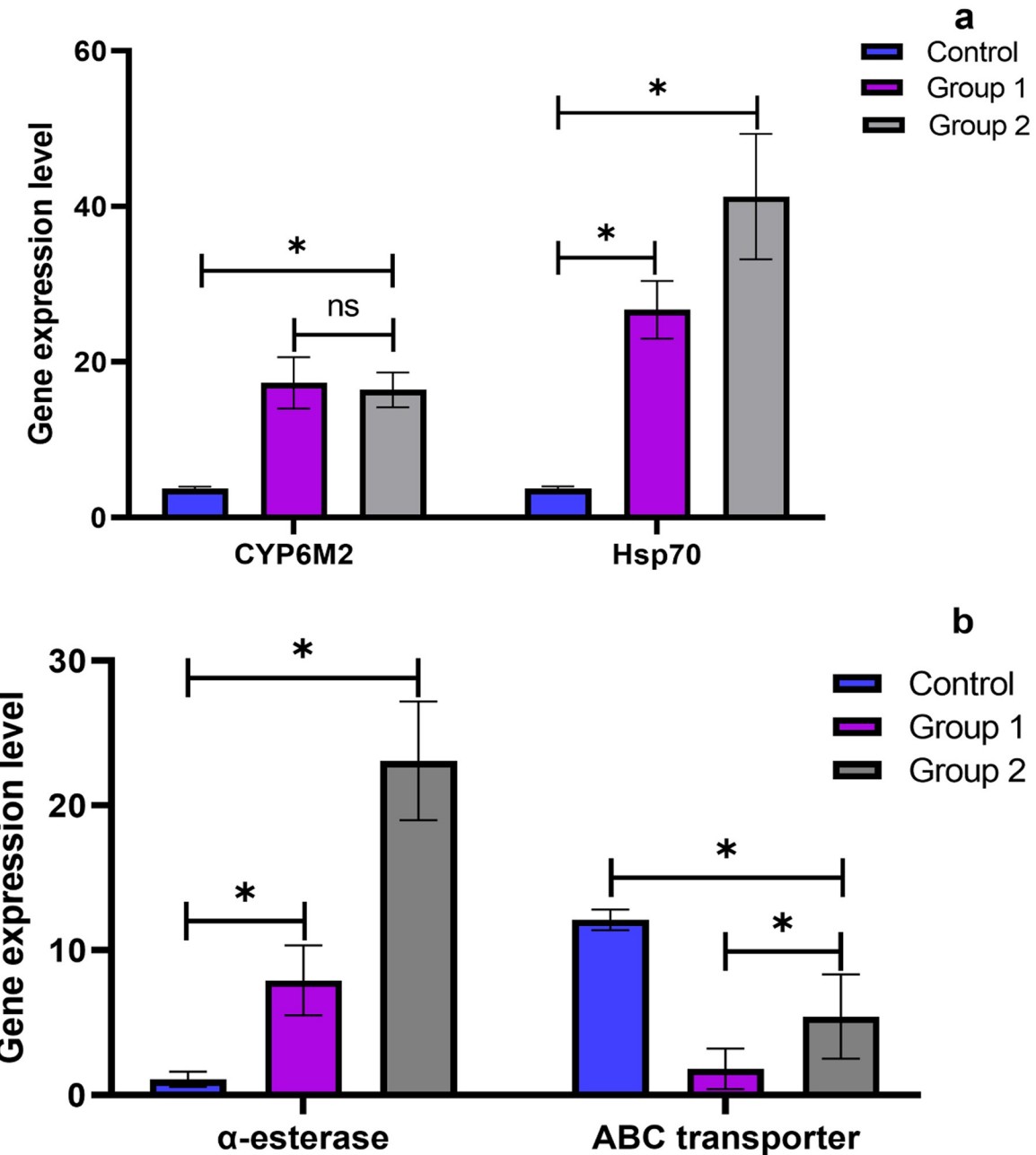

**Fig 6. Expression profile of *CYP6M2*, *Hsp70*, α-esterase and *ABC transporter* genes in larvae of *An. gambiae (s.s)*.** (A): larvae treated with methanol (Group 1) and aqueous (Group 2) showed significant expression of *CYP6M2*, *Hsp70* as compared to the control. (B): α-esterase was upregulated in larvae treated with both aqueous and methanol extracts while *ABC transporter* was significantly downregulated. * Values are significantly different at $p < 0.05$. ns: not significant.

## Modulation of antioxidant enzymes

The differential modulation of antioxidant enzymes in the larvae of *An. gambiae (s.s.)* by extracts of *O. tenuiflorum* is illustrated in Table 3 and Fig 7. The levels of SOD, CAT and GSH increased in larvae exposed to aqueous extract as compared to untreated larvae ($p < 0.05$). In contrast, GPx activity was significantly reduced in larvae exposed to 50, 125 and 250 mg/L

**Table 3. Antioxidant enzymes status of *An. gambiae (s.s.)* larvae exposed to *O. tenuiflorum* extracts.**

| *O. tenuiflorum* | Group (mg/L) | Antioxidant enzyme (U/mg protein) | | | |
|---|---|---|---|---|---|
| | | SOD | CAT | GSH | GPx |
| Aqueous extract | Control | 45.33±3.25[a] | 0.42±0.01[a] | 47.37±0.14[a] | 51.99±0.37[a] |
| | 12.5 | 56.92±0.66[b] | 0.45±0.00[b] | 59.10±0.38[b] | 51.85±0.80[a] |
| | 25 | 56.12±0.79[b] | 0.51±0.02[c] | 67.10±1.24[cd] | 55.40±0.95[b] |
| | 50 | 65.76±1.21[c] | 0.55±0.01[d] | 73.27±1.97[e] | 36.76±3.83[c] |
| | 125 | 67.47±0.68[cd] | 0.57±0.01[d] | 64.76±3.69[c] | 46.88±0.00[d] |
| | 250 | 70.05±0.74[d] | 0.46±0.00[b] | 71.32±0.80[de] | 34.47±0.59[c] |
| Methanol extract | Control | 44.10±2.38[a] | 0.46±0.02[a] | 56.01±1.97[a] | 44.99±2.67[a] |
| | 12.5 | 48.53±0.59[b] | 0.54±0.03[b] | 46.42±1.62[bc] | 54.06±0.69[bc] |
| | 25 | 55.12±0.31[c] | 0.48±0.00[a] | 43.36±1.40[b] | 58.19±0.89[d] |
| | 50 | 48.80±0.71[b] | 0.58±0.02[b] | 52.66±2.80[ac] | 51.38±1.12[b] |
| | 125 | 62.61±1.87[d] | 0.45±0.03[a] | 49.03±4.67[bc] | 55.28±1.74[cd] |
| | 250 | 58.95±0.52[e] | 0.45±0.00[a] | 56.44±1.36[a] | 55.26±0.42[cd] |

Values were expressed as mean± standard error of the mean of triplicates. Different superscripts ([a,b,c,d,e]) down the column indicate significant difference at $p < 0.05$.

concentrations of *O. tenuiflorum*. Similarly, in larvae exposed to different concentrations of methanol extracts of *O. tenuiflorum*, SOD, CAT and GPx were elevated as compared to the control ($p < 0.05$). Notably, GSH concentration was reduced in larvae exposed to 12.5, 25 and 250 mg/L of methanol extracts of *O. tenuiflorum*.

## Discussion

Mosquito management has relied on the use of various vector control strategies, mainly synthetic insecticides, to reduce incidences of mosquito-borne diseases. Due to the challenges associated with the use of synthetic chemicals, the use of eco-friendly and conveniently accessible plant-based alternatives has recently been proposed as a replacement [39]. Secondary metabolites found in natural products derived from plant extracts have already been shown to induce stress, alter larval development, and delay adult emergence [18, 24]. However, there is still a lack of understanding of the toxicity mechanisms of these plant extracts, particularly at the molecular level, which has limited their potential for absolute exploration as ecofriendly

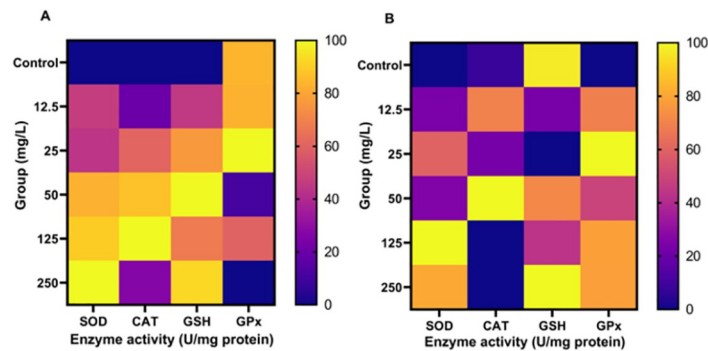

**Fig 7.** Heatmap showing normalized data of antioxidant enzymes activity in *An. gambiae (s.s.)* exposed to aqueous extract (A) and methanol extract (B) of *O. tenuiflorum*. Activity of antioxidant enzymes is normalized in percentage (%). SOD: superoxide dismutase; GSH: reduced glutathione; CAT: catalase and GPx: glutathione peroxidase.

plant-based alternatives. In this study, the effect of sublethal doses of *O. tenuiflorum* extracts on *An. gambiae* (*s.s.*) larvae and the putative molecular mechanisms by which the phytoactive extracts induce stress were investigated.

Plant-based biolarvicides are extremely beneficial due to the multimodal mechanism of action of their bioactive components. The aqueous extract of *O. tenuiflorum* analyzed in this study showed significant peaks of trans-β-ocimene, 6-methyl-2-methylene-6-(4-methyl-3-pentenyl)-bicyclo[3.1.1]heptane, 1-(1,5-dimethyl-4-hexenyl)-4-methylbenzene, 5-(1,5-dimethyl-4-hexenyl)-2-methyl-1,3-cyclohexadiene, cis-α-bisabolene, 3,7-dimethyl-1,3,6-octatriene, p-cymene, α-farnesene, hexadecyl-oxirane, octadecanoic acid, and cis,cis,cis-7,10,13-hexadecatrienal. Prior studies have documented the larvicidal, insecticidal, and pesticidal effects of these bioactive compounds [25–29, 31–34]. In the same way, the aqueous extracts contained significant amounts of o-cymene, 1,3,5-trimethyl-cyclohexane, 4-methoxy-benzaldehyde, and α-methyl-benzenemethanol. These compounds have been reported to have larvicidal and insecticidal properties [35–38]. These compounds possess a multifaceted larvicidal mechanism that reduces the probability of resistance development often observed with single active compounds, as they operate either individually or synergistically. In addition, the higher proportions of p-cymene (4.04%), hexadecyl-oxirane (4.11%), cis,cis,cis-7,10,13-hexadecatrienal (5.02%), trans-β-ocimene (5.75%), 3,7-dimethyl-1,3,6-octatriene (7.43%), octadecanoic acid (7.46%) in the aqueous extract and 1,3,5-trimethyl-cyclohexane (7.10%), 4-methoxy-benzaldehyde (44.80%) in methanol extract likely played a significant role in the observed toxicity against the larvae of *An. gambiae* (*s.s*).

The treatment of *An. gambiae* (*s.s.*) larvae with varied concentrations of methanol and aqueous extracts of *O. tenuiflorum* resulted in varying larvicidal potentials. The larvicidal mortality of the methanol extract at 24, 48 and 72 h were found to be 21.1%, 27.8% and 36.1% respectively, which was relatively lower as compared to the aqueous extracts with 26.1%, 36.1%, 45% at 24, 48, and 72 h, respectively. This shows a moderate larvicidal efficacy of aqueous and methanol extracts of *O. tenuiflorum* against *An. gambiae* (*s.s.*) larvae. These findings are consistent with findings by Ragavendran [16], who discovered that *O. tenuiflorum* had a moderate larvicidal and adulticidal effect on *Culex gelidus* and *Culex quinquefasciatus*. Other studies have found that plant extracts cause toxicity by extending larval development, reducing pupation and subsequent adult emergence rates, and decreasing egg output [18, 40, 41].

The molecular mechanisms behind the observed physiological adverse effects of aqueous and methanol extracts of *O. tenuiflorum* on *An. gambiae* (*s.s.*) larvae were also investigated. Insects have been observed to react to stress and eliminate harmful substances by increasing the production of stress proteins, particularly heat shock proteins. This leads to an increase in the activity of enzymes responsible for detoxification and a decrease in the levels of toxic substances within the cells. These processes are facilitated by specific integral membrane proteins [6, 11, 42]. The excessive production of detoxifying enzymes, such as cytochrome P450s and esterase, leads to the breakdown of xenobiotics, which in turn induces the development of tolerance to toxic substances.

In contrast to the untreated group, the application of *O. tenuiflorum* aqueous and methanol extracts resulted in a considerable increase in *CYP6M2* overexpression. The studies conducted by Vivekanandhan et al. [5] and Mitchell et al. [43] revealed a positive correlation between elevated levels of *CYP6M2* expression and resistance to DDT in *An. gambiae* (*s.s*), as well as resistance to temphos in *An. stephensi*. In addition, the aqueous and methanol extracts exhibited modest variations in the expression level of *CYP6M2* in *An. gambiae* (*s.s*). The extracts seem to stimulate the expression of *CYP6M2* in *An. gambiae* (*s.s.*) larvae in a comparable manner.

Insects' metabolism of toxic substances is significantly influenced by enzymes from the esterase family [44]. David et al. [6] showed that mosquitoes acquire resistance to pyrethroids

through the overexpression of their *α-esterase* gene. The larvae of *An. gambiae* (*s.s.*) exposed to aqueous and methanol extracts of *O. tenuiflorum* exhibited a significantly elevated level of *α-esterase* expression in comparison to the control group. Exposure of larvae to toxicants indicates their engagement in a comprehensive detoxification process. In addition, the aqueous extract of *An. gambiae* (*s.s.*) resulted in a higher expression of *α-esterase* compared to the methanol extract. Based on this information, it was shown that the larvae of *An. gambiae* (*s.s.*) exposed to methanol extracts displayed less toxicant metabolizing attributes compared to larvae treated with aqueous extracts of *O. tenuiflorum*. Additionally, α-esterase enzyme plays a significant role in the detoxification process in these mosquitoes. The detoxification process may have similarities to earlier research findings that suggest α-esterase is responsible for metabolic resistance in insects through the hydrolysis of synthetic chemicals [45, 46]. The findings indicate that the aqueous extract induced superior detoxifying activities in *An. gambiae* (*s.s.*) larvae in comparison to the methanol extract, despite both extracts having a moderate level of toxicity. Presumably, this is a result of the excessive expression of *CYP6M2* and *α-esterase* genes.

Insects produce heat shock proteins in reaction to many stressors such as high temperatures, lack of water, and exposure to foreign substances [11, 47, 48]. These proteins serve as universal molecular chaperones, ensuring the stability and functionality of physiological proteins. Insects' response to stressful conditions has been associated with various heat shock proteins, such as Hsp20, Hsp70, and Hsp90. Multiple studies have demonstrated that the expression of the *Hsp20* and *Hsp90* genes is significantly elevated in order to safeguard mosquito larvae from stress [11–13]. This study investigated the expression of *Hsp70*. Significantly elevated levels of *Hsp70* were detected in *An. gambiae* (*s.s.*) that were exposed to aqueous and methanol extracts of *O. tenuiflorum*, in comparison to the control group. This is consistent with the findings of Muema et al. [18], which demonstrated that *An. gambiae* (*s.s.*) exhibited elevated *Hsp70* expression as a defense mechanism against oxidative stress-induced toxicity. Importantly, mosquitoes treated with aqueous extract exhibited distinct expression patterns in comparison to those treated with methanol extracts. When *An. gambiae* (*s.s.*) is exposed to an aqueous extract instead of a methanol extract, there is a significant 15-fold increase in the expression of *Hsp70*. This indicates a heightened response to toxic stress. This discovery confirms the increased expression of *CYP6M2* and *α-esterase* genes in *An. gambiae* (*s.s.*) mosquitoes when exposed to the aqueous extract of *O. tenuiflorum*.

An effective cellular defense system against toxicants involves the expulsion of toxic substances from the cell, thereby decreasing their concentration within the cell [42]. Buss and Callaghan [49] state that ABC transporters serve as the primary defensive mechanism against many forms of xenobiotics. *An. stephensi* has been observed to upregulate these transporters as a defense mechanism against permethrin. Profiling the ABC transporter genes could be beneficial for vector and pest management strategies, as they exhibit a similar induction response to insecticide treatment in both mosquito larvae and adults. In this study, it was shown that larvae treated with methanol and aqueous extracts of *O. tenuiflorum* exhibited a significant reduction in ABC transporter activity, indicating a suppression effect. Hence, if the downregulation of the ABC transporter leads to alterations in the absorption, distribution, and elimination of medicines, it could potentially undermine the efficacy of the extracts. Also, further investigation has recorded the diverse categories of integral proteins present in ABC transporters. Mastrantonio et al. [50] provide thorough information on the differential expression and cellular defense function of several types of ABC transporters. For instance, ABCB3, ABCB4, and ABCC11 exhibited either no expression or down-regulation in *An. stephensi* when exposed to permethrin, while ABCG4, ABCB2, and ABCB6 were shown to be implicated in the defense against permethrin exposure. The lack of susceptibility of *An. gambiae* (*s.s.*) to methanol and

aqueous extracts of *O. tenuiflorum* suggests that the observed down-regulation of ABC transporters in this study may be attributed to the targeting of different types of ABC transporters.

Furthermore, to better ascertain the harmful characteristics, the antioxidant composition of *An. gambiae* (*s.s.*) larvae exposed to aqueous and methanol extracts of *O. tenuiflorum* was analyzed. Mosquitoes exhibit endogenous defense mechanisms against harmful substances, which involve both enzymatic and non-enzymatic defense systems. Specifically, antioxidant enzymes such as reduced GSH, GPx, and SOD help safeguard cells against harm caused by free radicals. Mosquitoes exhibit variable expressions of this group of enzymes in response to hazardous substances, as indicated by studies conducted by Aremu et al. [14] and Kazek et al. [51]. The results indicate that the activities of antioxidant enzymes are influenced to different degrees by varying concentrations of harmful substances. When comparing untreated larvae to those subjected to an aqueous extract of *O. tenuiflorum* containing 12.5–250 mg/L, the latter group showed a significant increase in GSH contents. Evidence suggests that the hazardous substances undergo a heightened conjugation process with glutathione, resulting in their neutralization and reduced level of harm. Consistent with the research conducted by Yildirim and Yaman [52] and Azeez et al. [53], an increase in GSH concentration suggests the activation of protective mechanisms against toxicity. *An. gambiae* (*s.s.*) larvae subjected to methanol extract doses of 12, 25, and 125 mg/L exhibited reduced GSH concentrations compared to larvae treated with different dosages. This is consistent with the findings of Aremu et al. [24], who discovered that the activity of antioxidant enzymes in *Cx. quinquefasciatus* mosquitoes was influenced in different ways by extracts of *Azadirachta indica* nanoparticles at specific concentrations. In addition, the activity of GPx in *An. gambiae* (*s.s.*) larvae increased at all doses of the methanol extract that were examined. However, a concentration as low as 25 mg/L of the aqueous extract showed a similar impact. The observed increase in GPx activity may be attributed to the toxicity of the extracts, resulting in an elevated degradation of organic hydroperoxides, such as hydrogen peroxide. In most cases, *An. gambiae* (*s.s.*) larvae treated with both aqueous and methanol extracts exhibited an increase in CAT and SOD activities, which is an additional benefit. The functional presence of SOD and CAT in larvae of *An. gambiae* (*s.s.*) is shown by their ability to facilitate the breakdown of the superoxide anion into oxygen and hydrogen peroxide, and then convert the hydrogen peroxide into water and oxygen.

## Conclusions

The findings suggest that the detoxification response of *An. gambiae* (*s.s.*) to phytoactive compounds in *O. tenuiflorum* is similar to that of other synthetic pesticides. The overexpression of *CYP6M2*, *Hsp70*, and *α-esterase*, coupled with increased levels of SOD, CAT, and GSH, were responsible for the moderate toxicity observed in *An. gambiae* (*s.s.*) exposed to aqueous and methanol extracts of *O. tenuiflorum*. A possible indication of toxicity is the down-regulation of the *ABC transporter*. These findings support the notion that exposure to xenobiotics in *An. gambiae* (*s.s.*) larvae triggers multifaceted and intricate transcriptomic and antioxidant responses.

## Supporting information

**S1 File. Data for the determination of mean percentage mortality at 24, 48 and 72 h of larvae of *An. gambiae (s.s.)* treated with extracts of *O. tenuiflorum*.**
(XLSX)

**S2 File. Data for the plot on survival rates of larvae exposed to different concentrations of aqueous (A) and methanol (B) extracts of *O. tenuiflorum*.**
(XLSX)

**S3 File. Data for the plot on expression profile of CYP6M2, Hsp70, α-esterase and ABC transporter genes in larvae of *An. gambiae* (*s.s.*).**
(XLSX)

**S4 File. Data for the Antioxidant enzymes status of *An. gambiae* (*s.s.*) larvae exposed to *O. tenuiflorum* extracts.**
(XLSX)

**S5 File. Data for plot on heatmap showing normalized antioxidant enzymes activity in *An. gambiae* (*s.s.*) exposed to aqueous extract (A) and methanol extract (B) of *O. tenuiflorum*.**
(XLSX)

## Acknowledgments

The authors would like to acknowledge the facilities provided by the Department of Biochemistry, Osun State University, Nigeria. We extend our special gratitude to Dr. Anush Chiappino-Pepe for her invaluable support in ensuring the seamless dissemination of our research within the scientific community.

## Author Contributions

**Conceptualization:** Harun K. Aremu.

**Data curation:** Harun K. Aremu, Christianah A. Dare.

**Formal analysis:** Harun K. Aremu, Idris A. Adekale, Luqmon A. Azeez.

**Investigation:** Harun K. Aremu, Idris A. Adekale, Bukunmi D. Adetunji.

**Methodology:** Harun K. Aremu, Christianah A. Dare, Idris A. Adekale, Bukunmi D. Adetunji.

**Project administration:** Harun K. Aremu.

**Resources:** Olu I. Oyewole.

**Supervision:** Dickson A. Musa, Luqmon A. Azeez, Olu I. Oyewole.

**Writing – original draft:** Harun K. Aremu.

**Writing – review & editing:** Christianah A. Dare, Idris A. Adekale, Dickson A. Musa, Luqmon A. Azeez, Olu I. Oyewole.

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
