## [Decision Letter · Decision Letter 0]

1 Mar 2024

PONE-D-24-03688Phytomediated stress modulates antioxidant status, induces overexpression of CYP6M2, Hsp70, α-esterase, and suppresses the ABC transporter in Anopheles gambiae (sensu stricto) exposed to Ocimum tenuiflorum extractsPLOS ONE

Dear Dr. Aremu,

Thank you for submitting your manuscript to PLOS ONE. After careful consideration, we feel that it has merit but does not fully meet PLOS ONE’s publication criteria as it currently stands. Therefore, we invite you to submit a revised version of the manuscript that addresses the points raised during the review process.

**ACADEMIC EDITOR: **I made a decision about your manuscript review each question made by the reviewers point by point. 

We look forward to receiving your revised manuscript.

Kind regards,

Mozaniel Santana de Oliveira, Ph.D

Academic Editor

PLOS ONE

Journal Requirements:

Reviewers' comments:

Reviewer's Responses to Questions

**Comments to the Author**

1. Is the manuscript technically sound, and do the data support the conclusions?

Reviewer #1: Yes

Reviewer #2: Yes

2. Has the statistical analysis been performed appropriately and rigorously? 

Reviewer #1: Yes

Reviewer #2: No

3. Have the authors made all data underlying the findings in their manuscript fully available?

Reviewer #1: Yes

Reviewer #2: Yes

4. Is the manuscript presented in an intelligible fashion and written in standard English?

Reviewer #1: Yes

Reviewer #2: Yes

5. Review Comments to the Author

Reviewer #1: The authors investigate the larvicidal effects and putative molecular mechanisms of Ocimum tenuiflorum extracts against the malaria vector Anopheles gambiae. This research is driven by the need for environmentally-safe mosquito control alternatives to replace synthetic insecticides. The topic is relevant and the paper is within the scope of the journal PLOS One. Overall, the manuscript is clearly written and the research approach is sound. However, there are a few areas that need strengthening:

1- The introduction could provide more background on the specific phytocompounds found in O. tenuiflorum that have known bioactivity and relate this to the rationale for testing this plant.

2- The sample size used in larval bioassays is quite small at n=20. Increasing to at least n=30 per treatment would give more statistical power to detect differences between doses and extract types.

3- The toxicity mechanisms focus only on gene expression changes. Incorporating measurements of enzyme activity would significantly strengthen the link between transcript and protein induction.

4- Expanding the transcriptomic analysis to include other detoxification and stress response genes could provide greater coverage of the molecular pathways involved. ABC transporters alone give limited mechanistic insight.

5- Assessing chronic toxicity over the full larval development would better reveal sub-lethal impacts compared to acute toxicity alone.

This paper presents a worthwhile study on plant-based mosquito control, but lacks sufficient depth in some areas. Addressing the above points would significantly improve the quality of the results and strengthen the conclusions that could be drawn from this research. With moderate revisions to expand the scope, this work would represent a solid contribution to the field.

Reviewer #2: The research adds to a large body of literature on the role of natural products on malaria vector control and therefore enriches the extant literature. However, there are minor comments that the authors should attend to:

1. The rationale for using two solvents in this study is not clear. The manner in which the discussion is rendered suggests that the authors were clear in their mind why they used the two solvents but this rationale must be provided to readers.

2. L180: What is meant by n=3 is not clear.

3. Results presented in Figure is rather silent on the doses yet there appear to be differences accounted for by differences in doses.

4. The distinction between Group 1 and Group 2 is first done in the Results sections. One would argue that it is helpful to do so in the methods section.

5. It will be helpful to spell out, under notes to tables and figures, any abbreviations in such visual aids.

6. How survival rates was determined is not clear. These details are needed under data analysis.

7. The interpretations of significant differences as reported in L333-336 and L370-372 are not clear. For example, what is the explanation for differences in toxicity between Group 1 and Group 2. I find that the lack of clear interpretation of the result to be a major weakness in this manuscript.

6. PLOS authors have the option to publish the peer review history of their article (what does this mean?). If published, this will include your full peer review and any attached files.

Reviewer #1: **Yes: **Prof. Dr. Amal Hassan Egyptian Atomic Energy Authority

Reviewer #2: No

---

## [Author Response · Author response to Decision Letter 0]

13 Mar 2024

Response to the Reviewers’ comments

Manuscript Number: PONE-D-24-03688

Article Title: Phytomediated stress modulates antioxidant status, induces overexpression of CYP6M2, Hsp70, α-esterase, and suppresses the ABC transporter in Anopheles gambiae (sensu stricto) exposed to Ocimum tenuiflorum extracts

Journal Title: PLOS ONE

The authors express their gratitude to the editor and thank the anonymous reviewers for their thorough and valuable review, suggestions, and comments on how to improve the paper's quality and accurate reporting of data. We have taken into account all of the feedback we have received in the amended manuscript.

All corrections are highlighted in red

Academic editor’s comments

Comment 1: Please ensure that your manuscript meets PLOS ONE's style requirements, including those for file naming. 

Response

- Thank you for this clarification. The File names have been carefully checked and corrected.

Comment 2: In your Methods section, please provide additional information regarding the permits you obtained for the work. Please ensure you have included the full name of the authority that approved the field site access and, if no permits were required, a brief statement explaining why.

Response

- This has been included in the subsection “Ethical statement” in Line 107 – 110

Comment 3: Submission of minimal data set

Response

- We have added all raw data used to build Graphs and Tables as Supporting Information file (S1_File.xlsx – S5_File.xlsx).

Reviewer #1

Comment 1: The introduction could provide more background on the specific phytocompounds found in O. tenuiflorum that have known bioactivity and relate this to the rationale for testing this plant.

Response

- This has been carefully checked and included in the introduction in Line 86 – 90. 

Comment 2: The sample size used in larval bioassays is quite small at n=20. Increasing to at least n=30 per treatment would give more statistical power to detect differences between doses and extract types.

Response: 

- Thank you for your valuable feedback. We acknowledge the potential benefits of increasing the sample size to enhance statistical power in detecting differences between doses and extract types in our larval bioassays. While we understand that a larger sample size, such as n=30 per treatment, could provide more robust statistical analysis, there were practical constraints that influenced our decision to use a sample size of n=20.

Firstly, our study was conducted under limited resource availability, including time and budget constraints. As such, increasing the sample size to n=30 per treatment would have necessitated additional resources beyond our current capacity. Additionally, we ensured that our chosen sample size of n=20 per treatment adhered to established standards within our field and was sufficient to detect meaningful differences based on preliminary power analyses. However, we acknowledge the importance of statistical power in research validity and will consider your suggestion for future investigations.

Comment 3: The toxicity mechanisms focus only on gene expression changes. Incorporating measurements of enzyme activity would significantly strengthen the link between transcript and protein induction.

Response

- We appreciate the insightful comment regarding the incorporation of measurements of enzyme activity to strengthen the link between transcript and protein induction in our study on toxicity mechanisms. While we agree that assessing enzyme activity would provide valuable complementary data, our decision to focus solely on gene expression changes was guided by several considerations.

Firstly, our primary objective was to investigate the transcriptional responses underlying toxicity mechanisms, as gene expression changes often serve as early indicators of cellular responses to toxic stimuli. By focusing on gene expression, we aimed to elucidate the molecular pathways and regulatory networks involved in the observed toxic effects.

Furthermore, conducting enzyme activity assays would have required additional resources and experimental procedures, which were not feasible within the scope of our study. Given the constraints of time and resources, we prioritized the transcriptomic analysis as it provided a comprehensive assessment of changes at the transcriptional level. We have also added the heatmap showing the activities of the antioxidant biomarkers.

Comment 4: Expanding the transcriptomic analysis to include other detoxification and stress response genes could provide greater coverage of the molecular pathways involved. ABC transporters alone give limited mechanistic insight.

Response

- Thank you for your valuable feedback. Aside from ABC transporters, other detoxification and stress response genes were also included in this study such as the Hsp70 (stress gene), CYP6M2 and α-esterase (detoxification genes). 

Comment 5: Assessing chronic toxicity over the full larval development would better reveal sub-lethal impacts compared to acute toxicity alone.

Response

- We appreciate the insightful comment. While we acknowledge the importance of chronic toxicity assessments, our study aimed to provide initial insights into the immediate effects of exposure to the tested substances on larval development and survival. Acute toxicity assessments allow for rapid evaluation of potential hazards and help prioritize substances for further investigation.

Reviewer #2 

Comment 1: The rationale for using two solvents in this study is not clear. The manner in which the discussion is rendered suggests that the authors were clear in their mind why they used the two solvents but this rationale must be provided to readers.

Response

- We appreciate this feedback. We have included the justification for the different solvents in the introduction (Line 93 - 94).

Comment 2: L180: What is meant by n=3 is not clear.

Response

- It means the mean and Standard deviation values were gotten from values in triplicate.

Comment 3: Results presented in Figure is rather silent on the doses yet there appear to be differences accounted for by differences in doses.

Response

- We appreciate this observation. The results presented in all the figures except gene expression showed the differences in doses. The gene expression was monitored only at the end of toxicant exposure after 72 hr for the two extracts; methanol (Group 1) and aqueous (Group 2). This was described in the section “Differential gene expression profile”.

Comment 2: The distinction between Group 1 and Group 2 is first done in the Results sections. One would argue that it is helpful to do so in the methods section.

Response

- We appreciate this observation. We have made the corrections in Line 166 – 168 of the Materials and methods section.

Comment 3: It will be helpful to spell out, under notes to tables and figures, any abbreviations in such visual aids.

Response

- We appreciate this observation. We have made the corrections in Line 166 – 168 of the Materials and methods section.

Comment 3: It will be helpful to spell out, under notes to tables and figures, any abbreviations in such visual aids.

Response

- This observation has been made.

Comment 3: How survival rates was determined is not clear. These details are needed under data analysis.

Response

- This survival analysis was carried out using the Kaplan-Meier survival analysis tool incorporated in the GraphPad Prism and has been included in the Statistical analysis subsection.

Comment 4: The interpretations of significant differences as reported in L333-336 and L370-372 are not clear. For example, what is the explanation for differences in toxicity between Group 1 and Group 2.

Response

- We appreciate this observation. We have rephrased the sentence for better clarity. In L333-336, we described the overexpression of the α-esterase in the larvae of Anopheles gambiae exposed to both aqueous and methanol extracts. From our results, we observed that there was more overexpression of α-esterase in An. gambiae treated with aqueous extract as compared to the methanol extract. We believed this might have played a role in the perceived moderate mortality of the extract. This is what was been discussed.

---

## [Decision Letter · Decision Letter 1]

27 Mar 2024

PONE-D-24-03688R1Phytomediated stress modulates antioxidant status, induces overexpression of CYP6M2, Hsp70, α-esterase, and suppresses the ABC transporter in Anopheles gambiae (sensu stricto) exposed to Ocimum tenuiflorum extractsPLOS ONE

Dear Dr. Aremu,

Thank you for submitting your manuscript to PLOS ONE. After careful consideration, we feel that it has merit but does not fully meet PLOS ONE’s publication criteria as it currently stands. Therefore, we invite you to submit a revised version of the manuscript that addresses the points raised during the review process.

**ACADEMIC EDITOR: **Dear author, I have received the reviewers' report, I ask you to review it point by point. 

We look forward to receiving your revised manuscript.

Kind regards,

Mozaniel Santana de Oliveira, Ph.D

Academic Editor

PLOS ONE

Journal Requirements:

Reviewers' comments:

Reviewer's Responses to Questions

**Comments to the Author**

1. If the authors have adequately addressed your comments raised in a previous round of review and you feel that this manuscript is now acceptable for publication, you may indicate that here to bypass the “Comments to the Author” section, enter your conflict of interest statement in the “Confidential to Editor” section, and submit your "Accept" recommendation.

Reviewer #1: All comments have been addressed

2. Is the manuscript technically sound, and do the data support the conclusions?

Reviewer #1: Yes

3. Has the statistical analysis been performed appropriately and rigorously? 

Reviewer #1: Yes

4. Have the authors made all data underlying the findings in their manuscript fully available?

Reviewer #1: Yes

5. Is the manuscript presented in an intelligible fashion and written in standard English?

Reviewer #1: No

6. Review Comments to the Author

Reviewer #1: Dear authors,

The manuscript requires language revision. There are several instances where the language needs improvement to ensure clarity and coherence. It is recommended to review the manuscript for grammar, sentence structure, and overall readability. Additionally, attention should be given to ensuring that the scientific terminology is accurately used and effectively communicated. A thorough language revision will greatly enhance the manuscript's quality and readability.

7. PLOS authors have the option to publish the peer review history of their article (what does this mean?). If published, this will include your full peer review and any attached files.

Reviewer #1: **Yes: **Professor Dr. Amal Hassan

---

## [Author Response · Author response to Decision Letter 1]

27 Mar 2024

Response to the Reviewers’ comments

Manuscript Number: PONE-D-24-03688R1

Article Title: Phytomediated stress modulates antioxidant status, induces overexpression of CYP6M2, Hsp70, α-esterase, and suppresses the ABC transporter in Anopheles gambiae (sensu stricto) exposed to Ocimum tenuiflorum extracts

Journal Title: PLOS ONE

The authors express their gratitude to the editor and thank the anonymous reviewers for their thorough and valuable review, suggestions, and comments on how to improve the paper's quality and accurate reporting of data. We have taken into account all of the feedback we have received in the amended manuscript.

All corrections are highlighted in red

Academic editor’s comments

Comment 1: Please review your reference list to ensure that it is complete and correct. If you have cited papers that have been retracted, please include the rationale for doing so in the manuscript text, or remove these references and replace them with relevant current references. Any changes to the reference list should be mentioned in the rebuttal letter that accompanies your revised manuscript. If you need to cite a retracted article, indicate the article’s retracted status in the References list and also include a citation and full reference for the retraction notice.

Response

- Thank you for this feedback and comment. We have made the following corrections to the references;

i. The citation, Kamaraj and Rahuman was removed because it was not part of the reference list and replaced with the accurate citation of Ragavendran [16]. (Line 317).

ii. Reference [39] which is the first citation in the discussion was changed to reference [24] in order to follow the chronological order of referencing as stipulated in the authors guideline. Therefore, this change affected the numbering of citations between 24 – 39. Consequently, previously citations of [24], [25], [26], [27], [28], [29], [30], [31], [32], [33], [34], [35], [36], [37], [38] and [39] were changed to [25], [26], [27], [28], [29], [30], [31], [32], [33], [34], [35], [36], [37], [38], [39], [40] respectively. These corrections were also equally done in the Reference list.

- Also, we have checked the references list and none of the cited papers have been retracted.

Reviewer #1

Comment 1: The manuscript requires language revision.

Response

- We have subjected the manuscript to Grammarly checker to further present the sentences with better clarity and coherence.

---

## [Decision Letter · Decision Letter 2]

9 Apr 2024

Phytomediated stress modulates antioxidant status, induces overexpression of CYP6M2, Hsp70, α-esterase, and suppresses the ABC transporter in Anopheles gambiae (sensu stricto) exposed to Ocimum tenuiflorum extracts

PONE-D-24-03688R2

Dear Dr. Aremu,

We’re pleased to inform you that your manuscript has been judged scientifically suitable for publication and will be formally accepted for publication once it meets all outstanding technical requirements.

Kind regards,

Mozaniel Santana de Oliveira, Ph.D

Academic Editor

PLOS ONE

Additional Editor Comments (optional):

Reviewers' comments:

Reviewer's Responses to Questions

**Comments to the Author**

1. If the authors have adequately addressed your comments raised in a previous round of review and you feel that this manuscript is now acceptable for publication, you may indicate that here to bypass the “Comments to the Author” section, enter your conflict of interest statement in the “Confidential to Editor” section, and submit your "Accept" recommendation.

Reviewer #1: All comments have been addressed

2. Is the manuscript technically sound, and do the data support the conclusions?

Reviewer #1: Yes

3. Has the statistical analysis been performed appropriately and rigorously? 

Reviewer #1: Yes

4. Have the authors made all data underlying the findings in their manuscript fully available?

Reviewer #1: Yes

5. Is the manuscript presented in an intelligible fashion and written in standard English?

Reviewer #1: Yes

6. Review Comments to the Author

Reviewer #1: Dear Authors,

I am pleased to inform you that your manuscript has been accepted for publication in its current form.

7. PLOS authors have the option to publish the peer review history of their article (what does this mean?). If published, this will include your full peer review and any attached files.

Reviewer #1: **Yes: **Prof. Dr. Amal Hassan, Radioisotopes Department, Egyptian Atomic Energy Authority
